# **KNOWDA**: ALL-IN-ONE KNOWLEDGE MIXTURE MODEL FOR DATA AUGMENTATION IN LOW-RESOURCE NLP TASKS

**Yufei Wang** [1]*, **Jiayi Zheng** [2], **Can Xu** [3], **Xiubo Geng** [3], **Tao Shen** [3], **Chongyang Tao** [3], **Daxin Jiang** [3]†

Macquarie University, Sydney, Australia[1], Peking University, Beijing, China[2]
Microsoft Corporation, Beijing, China[3]
`yufei.wang@students.mq.edu.au, zhengjiayi980@126.com`
`{caxu,xiubo.geng,shentao,chongyang.tao,djiang}@microsoft.com`

## ABSTRACT

This paper focuses on the data augmentation for low-resource NLP tasks where the training set is limited. The existing solutions either leverage task-independent heuristic rules (e.g., Synonym Replacement) or fine-tune general-purpose pre-trained language models (e.g., GPT2) using the limited training instances to produce new synthetic data. Consequently, they have trivial *task-specific knowledge* and are limited to yielding *low-quality* synthetic data. To combat this issue, we propose **Know**ledge Mixture **D**ata **A**ugmentation Model (**KnowDA**) which is an Seq2Seq language model pretrained on a mixture of diverse NLP tasks under a novel framework of **K**nowledge **M**ixture **T**raining (KoMT). The goal of KoMT is to condense diverse NLP task-specific knowledge into the single **KnowDA** model (i.e., all-in-one) such that **KnowDA** could utilize these knowledge to quickly grasp the inherent synthesis law of the target task through limited training instances. Specifically, KoMT reformulates input examples from various heterogeneous NLP tasks into a unified text-to-text format, and employs denoising training objectives in different granularity to learn to reconstruct partial or complete samples. To the best of our knowledge, we are the first attempt to apply 100+ NLP multi-task training for data augmentation. Extensive experiments show that i) the synthetic data produced by **KnowDA** successfully improves performance of the strong pre-trained language models (i.e., Bert, ALBert and Deberta) by a large margin on the low-resource NLP benchmark FewGLUE, CoNLL'03 and WikiAnn; ii) **KnowDA** successfully transfer the task knowledge to NLP tasks whose types are seen and unseen in KoMT. [1]

## 1 INTRODUCTION

Neural NLP models require extensive supervised training data to achieve superior performance (Bowman et al., 2015). However, due to the enormous cost of annotating data, developers could only use limited labeled data for training in common real-world uses of neural NLP models. This problem has attracted considerable attention recently. Many researchers (Kumar et al., 2020; Wang et al., 2022; Zhou et al., 2021) resort to data augmentation techniques to generate more synthetic samples to boost the performance of low-resource NLP tasks.

The existing NLP data augmentation (DA) methods either leverage task-independent heuristic rules, such as Synonym Replacement (Zhang et al., 2015) and Random Swap (Wei & Zou, 2019a), or fine-tune general-purpose pre-trained language models by using the handful training examples of target tasks, such as GPT2 (Radford et al., 2019) in LAMBADA (Anaby-Tavor et al., 2020) and T5 (Raffel et al., 2020) in PromDA (Wang et al., 2022), to produce new synthetic data. Consequently, these DA methods have trivial *target task knowledge* and are limited to yielding low-quality synthetic data (e.g., either irrelevant or extremely similar to the training data). In addition, these DA methods

---

*Work done during the internship at Microsoft STCA.
†Corresponding author: Daxin Jiang (djiang@microsoft.com).
[1]The source code is released in `https://github.com/GaryYufei/ICLR2023_KnowDA`.

are often applied to *simple* NLP tasks where the inputs only include single and short sentences. Recently, Zhou et al. (2021) demonstrate that these DA methods perform even worse on tasks with complicated structures (e.g., SuperGLUE). These issues prevent the existing DA methods from practical usage for various low-resource NLP tasks.

Motivated by this, in this paper, we propose **Know**ledge Mixture **D**ata **A**ugmentation Model (**KnowDA**) to tackle these two issues. To enrich the task knowledge for **KnowDA**, we propose *Knowledge Mixture Training* (KoMT) to represent various heterogeneous NLP tasks in a unified and scalable manner. Specifically, in KoMT, arbitrary NLP task instances are represented as the key-value list format where the key is typically a short phrase indicating the feature function and the value is a string representation of the feature content. We further employ the denoising training objectives in different granularity of the NLP task instances. That is, during KoMT, we randomly mask a subset of values (e.g., an input document or a question) and train **KnowDA** to reconstruct those masked ones. With the dynamic multi-granularity masking mechanism and the unified format, we successfully scale *Knowledge Mixture Training* (KoMT) of LM to about 100 NLP tasks without much human effort. Compared with previous attempts in unified multi-task learning works (e.g., T0 (Wei et al., 2022) and FLAN (Wei et al., 2022)), KoMT is more scalable and comprehensive because those works heavily rely on the human-crafted prompts and are only trained to improve the NLP task performance (i.e., only generating correct output labels). Furthermore, previous data augmentation methods only focus on *simple* NLP tasks, such as single-sentence classification. However, modern NLP tasks, such as NLI and QA, have much more complicated structures (i.e., multiple input text and long documents). To enable **KnowDA** to handle these NLP tasks with complicated structures, we propose a novel auto-regressive generation framework for **KnowDA**. At each step, we either fine-tune or directly use (i.e., zero-shot) a separate copy of **KnowDA** to generate a textual feature in an NLP instance. The feature generation order is based on *task-specific feature dependency*. As **KnowDA** is trained generate arbitrary input text features, we find it beneficial to generate long text in a zero-shot manner directly using the **KnowDA** checkpoint. Under this scenario, we control the outputs of **KnowDA** using relevant feature keys and full example demonstrations from the target tasks.

For evaluation, we conduct experiments on the challenging FewGLUE benchmark (Schick & Schütze, 2020) with 32 training examples. We also verify the effectiveness of **KnowDA** on two Sequence Labeling tasks, CoNLL'03 (Sang & De Meulder, 2003) and WikiAnn (Pan et al., 2017), whose task types are held-out during KoMT. **KnowDA** successfully outperforms recently proposed state-of-the-art data augmentation algorithms such as FlipDA and PromDA (Wang et al., 2022). We further compare the quality of generated synthetic data from **KnowDA** and FlipDA, confirming that **KnowDA** produces synthetic data with a higher level of diversity and better quality verified by humans.

The contributions of this paper are the following:(1) To the best of our knowledge, we are the first work to scale the number of tasks to 100+ in multitask pretraining for data augmentation; (2) We propose a novel multitask pre-training approach KoMT for data augmentation, resulting in a new pre-trained model, **KnowDA**; and (3) **KnowDA** outperforms state-of-the-art data augmentation methods on the low-resource setting of popular benchmarks FewGLUE, CoNLL'03, and WikiAnn.

## 2 METHOD

In this section, our setting is introduced in Sec. 2.1. The details of **KnowDA**, including the design of KoMT and the auto-regressive data augmentation procedure, are discussed in Sections 2.3 and 2.4.

### 2.1 DATA AUGMENTATION FOR LOW-RESOURCE NLP

In the low-resource NLP tasks, only a handful of labeled training data $\mathcal{T} = \{(x_i, y_i)\}_{i=1}^n$ are available. *Data Augmentation* generates synthetic data $\mathcal{T}_{Syn} = \{(\hat{x}_i, \hat{y}_i)\}_{i=1}^m$ from the original labeled training data $\mathcal{T}$ using language models, where $m$ is allowed to be much larger than $n$. The goal is that NLP models trained using $\mathcal{T} \cup \mathcal{T}_{Syn}$ outperform the ones only trained using $\mathcal{T}$.

### 2.2 OVERVIEW OF **KNOWDA**

**KnowDA** is an encoder-decoder generative language model that generates task-relevant and diverse synthetic data *from scratch*. It is initialized from an existing pre-trained encoder-decoder language

model. We further apply *Knowledge Mixture Training* (Sec. 2.3) to inject diverse NLP task-specific knowledge into **KnowDA**. Finally, we fine-tune or directly use **KnowDA** to produce synthetic data to improve strong NLP baseline models in various low-resource NLP tasks (Sec. 2.4).

## 2.3 KNOWLEDGE MIXTURE TRAINING

To recap, previous data augmentation methods lack *task-specific knowledge*. Currently, the research community has accumulated a large amount of various human-generated datasets. The main motivation of **KnowDA** is to transfer the task knowledge in the existing large amount of NLP supervisions to the generated synthetic data for other NLP tasks. The resulting models should learn to generate task-related data examples merely from a handful training samples of the target task. However, existing artificially generated NLP tasks have a variety of complex structures (e.g., sentence pairs classification, multiple-choice QA) that makes learning a unified data generator quite challenging. We convert the generation of data examples from various heterogeneous NLP tasks into a unified text-to-text format and employ denoising objectives in different granularity to simulate the reconstruction of partial or whole instances. We call this training paradigm *Knowledge Mixture Training* (KoMT). We will elaborate on KoMT in detail from following aspects: task collection, unified text-to-text format, demonstration examples and the denoising objectives.

**Task Collection**   The success of KoMT is built upon a resource with a sufficient number of tasks, covering a diverse range of NLP applications. Similar to Ye et al. (2021), we select English monolingual datasets with open access in the Huggingface Datasets (Lhoest et al., 2021). The tasks in KoMT broadly belong to the following task families: Text Classification, Natural Language Inference, Reading comprehension, Question Answering, Summarization, and other NLP applications. Training examples from each task are sampled proportionate to the size of individual NLP dataset. Each NLP dataset contribute at most 300k instances during KoMT.

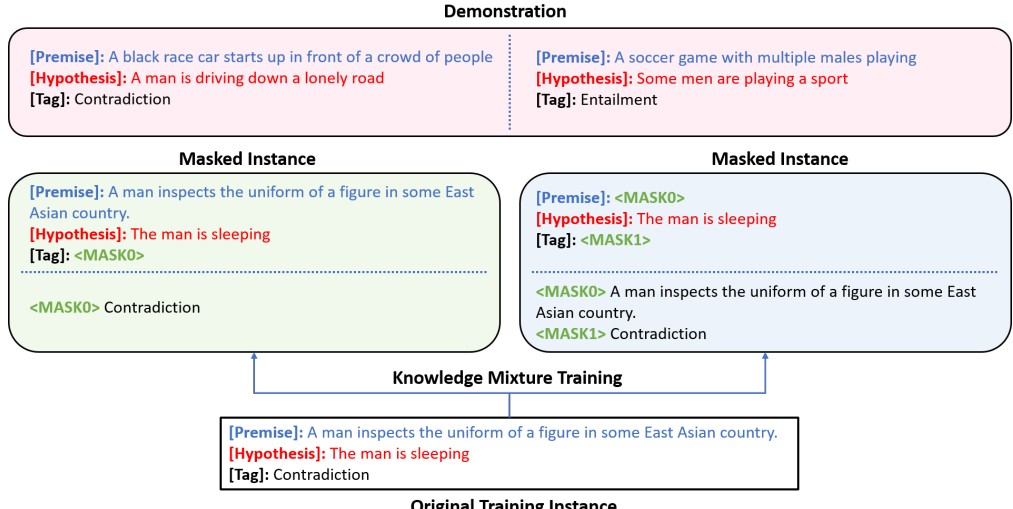

Figure 1: A running example of the masked training instances in the *Knowledge Mixture Training*. Keywords within the brackets are the task key.

**Unified Text-to-Text Format**   Although there are many NLP tasks with various structures, all of these tasks include one or more input features and an output result. For example, we have an input sentence/paragraph/document and an output label in the sentence classification tasks. Each feature, including the output result, could be described as a key-value pair. The key is typically a short phrase indicating the feature function, and the value is a string representation of the feature content. That is, given an instance $(x_i, y_i)$ from arbitrary NLP tasks, we could always represent it as a unified key-value list format (length $n$): $(x_i, y_i) = [(k_i^1, v_i^1), \cdots, (k_i^n, v_i^n)]$ where each $(k_i^j, v_i^j)$ either corresponds to an input feature or the output result $y_i$. Unlike previous works (Wei et al., 2022; Sanh et al., 2022) requiring exhausting human-crafted prompts to perform multi-task scaling, our

key-value design only needs the feature name in the original NLP dataset. Therefore, our proposed unified format could quickly extend KoMT to arbitrary tasks without much human effort.

**Denoising Objectives**    Based on the unified key-value list format, our denoising objective is defined as follows: given a training instance, we randomly mask K values in the key-value list, where number K is sampled from range $[1$ to $n]$ with equal probability and n is the total number of key-value pairs. Then, **KnowDA** is trained with the objective to reconstruct those masked values using the remaining information. In addition, we will prepend $L$ examples as demonstrations in the front of the input to **KnowDA** during KoMT, where $L$ is sampled from the range $[0$ to $m]$ with equal probability and $m$ is the hyperparameter of how many demonstrations to put at most. Figure 1 shows two masked training instances from the CB task (De Marneffe et al., 2019) where Premise and Hypothesis are input features and Tag is the output result. This unified format only requires simple cross-entropy loss, avoiding extra task-specific effort for loss design, loss scaling, or explicit gradient accumulation (Aghajanyan et al., 2021). This strategy trains **KnowDA** to reconstruct NLP task instances from multiple granularities, potentially satisfying different needs in the data augmentation tasks. Our dynamic random masking mechanism also encourages **KnowDA** to fully use the task-specific keys and demonstration examples. When most of the values are masked for prediction, they are the only reliable information for **KnowDA** to generate meaningful synthetic data. For example, in Figure 1, when both "Premise" and "Hypothesis" are masked, **KnowDA** could only obtain task-relevant information from keys and demonstrations.

**Demonstration Exemplars**    The unified task representations and denoising objectives allow **KnowDA** to encode various task knowledge in the model parameters, *implicitly*. When handling the low-resource target tasks that are dissimilar to the existing training tasks in KoMT, such implicit task knowledge transfer could become less effective. To better represent the semantics of the low-resource target tasks, we add a demonstration section in our text-to-text format like GPT-3. As shown in Figure 1, during KoMT, we randomly select a few instances for the demonstration section from the identical tasks and never take any value masking. To allow **KnowDA** to cope with a different number of demonstration exemplars, the number of selected instances is a random number between 0 and 16. We also discover that demonstration exemplars play an important role when **KnowDA** generates long text in a zero-shot fusion (Details See Sec. 2.4).

## 2.4    GENERATING SYNTHETIC DATA USING **KNOWDA**

After KoMT, **KnowDA** has already captured various NLP task knowledge. It could be further adapted to generate synthetic data for low-resource NLP tasks. As discussed in Sec. 2.3, a NLP training instance could be represented as a list of key-value pairs $[(k^1, v^1), \cdots, (k^n, v^n)]$ where the key $k^i$ is the feature name and $v^i$ is the textual content. To generate NLP training instances with arbitrary complicated structures from scratch, we propose an *auto-regressive* data augmentation framework. We first analyze the *task-specific feature dependency*, then generate each feature value sequentially.

**Task-specific Feature Dependency**    Different NLP tasks have their own dependencies for input features. These dependencies are mostly based on the annotation procedure of the target task. For example, as shown in Figure 2, in the task of Adversarial QA (Bartolo et al., 2020a), given a passage (i.e., context), a human annotator first produces a question, and then creates the corresponding answer span for that question. Given this annotation procedure, we generate context, question, and answer auto-regressively. To maintain a high diversity of the generated synthetic data in the low-resource setting, we train a different copy of **KnowDA** for generation at each stage. This is conceptually similar to the *Prompt ensemble* used in Wang et al. (2022). Note that the fine-tuning cost (i.e., training time) of **KnowDA** at each stage is relatively low. Practically, we only need to store the generated synthetic data and safely discard the fine-tuned **KnowDA** parameters. The detailed data augmentation process used in this paper can be found in Appendix A.5.

**Generating Text For Features**    For short $v^j$ (e.g., less than 250 tokens), directly fine-tuning **KnowDA** with the low-resource training data is a straightforward solution. However, it is tremendously difficult to handle long $v^j$. In our preliminary experiments, we find that fine-tuned **KnowDA** memorizes some textual spans from the training examples and injects these memorized textual spans into the generated text. As a result, the generated text is inconsistent across different sentences and of

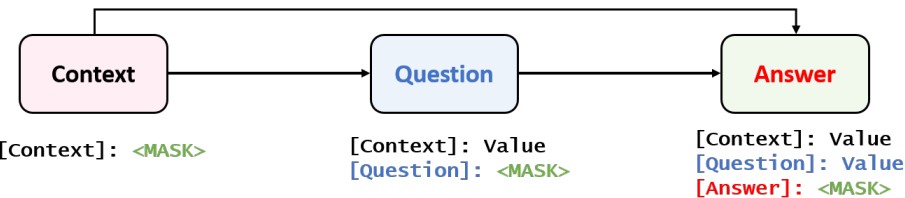

Figure 2: Feature Dependency in the task of Adversarial QA.

low quality. To avoid this fine-tuning issue, we directly use **KnowDA**, without further fine-tuning, to produce long text because **KnowDA** is pre-trained to generate many long texts. However, this could result in irrelevant outputs compared to the given handful of training examples in the target task. To combat this issue, we transfer task knowledge from the pre-training tasks in KoMT and the target task. Specifically, we select relevant feature keys based on the similarity (e.g., length and topic) between the target task and the pre-training tasks. To fully use the small set of training data in the target task, we randomly select a few of them and use them as full demonstration examples at the inputs. Both strategies could effectively control the contents of generated text, achieving satisfying data augmentation performance. We show the importance of these strategies in Sections 3.3.1 and 3.3.2.

## 3 EXPERIMENTS

In this paper, to show the effectiveness of **KnowDA**, we compare **KnowDA** with recently proposed state-of-the-art data augmentation algorithm FlipDA (Zhou et al., 2022) and PromDA (Wang et al., 2022) in Sections 3.1 and 3.2. We also present an in-depth analysis of **KnowDA** in Sec 3.3.

**Evaluation Benchmark**    Following PromDA and FlipDA, we conduct low-source experiments on the FewGLUE (Schick & Schütze, 2020), CoNLL'03 (Sang & De Meulder, 2003), and WikiAnn (Pan et al., 2017) benchmarks. For each benchmark, we run our experiments multiple times with different random seeds, templates and data splits and report the averaged results with the corresponding standard deviation. We write the standard deviation as the subscript of mean results (e.g., $80.0_{1.0}$ means the averaged performance is 80.0 with standard deviation 1.0). To avoid any form of data leakage, we exclude all above evaluation tasks from KoMT. We further filter out any training instances containing evaluation task input features (e.g., input sentences) from KoMT. Finally, 114 diverse NLP tasks (See Appendix A.3 for details) are selected for KoMT.

**KnowDA Details**    **KnowDA** is initialized from the T5-1.1-Large model (Raffel et al., 2020). We train **KnowDA** for 100k steps with a maximum sequence length of 512 and batch size 2048 in a Linux environment with $16 \times$ A100 GPU (40G). Fine-tuning **KnowDA** is carried out only using a single A100 GPU (40G). We use Adam as the optimizer to train **KnowDA**. More details see Appendix A.1.

### 3.1 DATA AUGMENTATION ON FEWGLUE

We use FewGLUE to evaluate **KnowDA**'s ability on low-resource data augmentation. Each challenging FewGLUE task only has 32 labeled instances.

**Setting** In this experiment, we largely follow the settings in FlipDA (Zhou et al., 2021), including the baseline models, the input templates and the random seeds. We compare **KnowDA** with **SR** (Zhang et al., 2015), **EDA** (Wei & Zou, 2019b), **T5-MLM** (Zhou et al., 2021), and **FlipDA** (Zhou et al., 2021). The PET templates and random seeds are identical to FlipDA. Tables 1 and 2 show the experiment results of FewGLUE. As FlipDA outperforms other baseline models by a large margin, we rerun the official code of FlipDA to compare its mean and standard deviation results with **KnowDA**. Results of baseline, **SR**, **EDA** and **T5-MLM** are taken from Zhou et al. (2021). More details see Appendix A.2.

**Result** Adding **KnowDA**'s synthetic data improves the baseline models in 7 out of 8 FewGLUE tasks, with an average improvement of 5.2 and 4.1 in the ALBERT and DeBERTa baseline models, respectively. On average, **KnowDA** outperforms previous state-of-the-art FlipDA by 1.6 and 2.2 on the ALBERT and DeBERTa models, respectively. **KnowDA** consistently exceeds FlipDA on 5 out

of 8 tasks, which include QA (BoolQ, MultiRC), NLI (RTE, CB), and Word Sense Disambiguation (WiC). This shows that KoMT is capable of transferring prior task-specific knowledge to *similar tasks* (i.e., QA, NLI), as well as *novel tasks* (i.e., Word Sense Disambiguation). Finally, **KnowDA** performs worse than FlipDA in the task of COPA and ReCoRD. This could be because **KnowDA** is unable to produce sufficient correct output labels for these two tasks.

Table 1: Performance of Baseline and **KnowDA** on FewGLUE with ALBERT-xxlarge-v2 model.

| Method | BOOLQ Acc. | RTE Acc. | CB. Acc./F1 | WiC Acc. | WSC Acc. | MultiRC EM/F1a | COPA Acc. | ReCoRD Acc./F1 | avg |
|---|---|---|---|---|---|---|---|---|---|
| Baseline | 72.5 | 61.4 | 82.7/74.8 | 51.3 | 77.0 | 33.0/74.6 | 88.3 | 86.2/86.8 | 71.2 |
| SR | 75.0 | 59.2 | 83.3/78.1 | 51.3 | 78.7 | 34.1/75.6 | 87.5 | 85.6/86.1 | 71.6 |
| EDA | 72.7 | 58.3 | 81.1/73.6 | 51.8 | 75.9 | 28.7/73.1 | 84.5 | 85.4/86.0 | 69.6 |
| T5-MLM | 73.9 | 62.3 | 83.5/75.0 | 51.1 | **79.2** | 33.8/74.1 | 87.3 | 85.2/85.7 | 71.5 |
| FlipDA | $77.0_{1.3}$ | $70.7_{2.6}$ | $86.3_{3.5}/82.5_{5.8}$ | $53.4_{1.0}$ | $78.7_{4.3}$ | $36.4_{2.1}/76.2_{1.1}$ | $\mathbf{89.2_{1.2}}$ | $\mathbf{86.0_{0.2}}/86.6_{0.2}$ | 74.8 |
| **KnowDA** | $\mathbf{78.2_{0.8}}$ | $\mathbf{78.7_{0.9}}$ | $\mathbf{89.6_{1.9}/85.4_{2.9}}$ | $\mathbf{55.9_{1.5}}$ | $77.9_{3.6}$ | $\mathbf{36.8_{1.1}/76.2_{0.7}}$ | $89.0_{2.1}$ | $85.9_{0.2}/\mathbf{86.6_{0.2}}$ | **76.4** |

Table 2: Performance of Baseline and **KnowDA** on FewGLUE with DeBERTa-v2-xxlarge model.

| Method | BOOLQ Acc. | RTE Acc. | CB. Acc./F1 | WiC Acc. | WSC Acc. | MultiRC EM/F1a | COPA Acc. | ReCoRD Acc./F1 | avg |
|---|---|---|---|---|---|---|---|---|---|
| Baseline | 78.3 | 82.0 | 85.4/79.3 | 58.7 | 80.1 | 40.4/78.1 | 87.7 | 90.2/90.8 | 77.4 |
| SR | 77.4 | 76.3 | 87.2/80.3 | 58.9 | 80.9 | 35.7/76.3 | 87.0 | 89.1/89.6 | 76.2 |
| EDA | 74.4 | 77.4 | 83.6/76.2 | 59.3 | 78.7 | 37.0/77.1 | 85.8 | 88.1/88.6 | 75.1 |
| T5-MLM | 77.4 | 81.2 | 83.0/73.7 | 60.7 | 82.4 | 35.0/75.0 | 88.2 | 89.7/90.3 | 76.7 |
| FlipDA | $81.6_{1.1}$ | $83.0_{1.3}$ | $88.1_{2.2}/86.1_{3.5}$ | $64.2_{1.4}$ | $77.6_{3.0}$ | $43.6_{1.5}/79.8_{0.8}$ | $\mathbf{86.5_{6.5}}$ | $\mathbf{90.7_{0.2}/91.3_{0.2}}$ | 79.3 |
| **KnowDA** | $\mathbf{83.9_{1.5}}$ | $\mathbf{86.8_{1.9}}$ | $\mathbf{92.1_{1.5}/89.2_{1.3}}$ | $\mathbf{66.5_{3.4}}$ | $79.5_{2.8}$ | $\mathbf{46.8_{2.5}/81.3_{0.7}}$ | $90.7_{5.0}$ | $89.7_{0.3}/90.3_{0.3}$ | **81.5** |

## 3.2 DATA AUGMENTATION FOR SEQUENCE LABELING TASKS

In Sec. 3.1, **KnowDA** shows success in handling NLP tasks with complicated structures from Super-GLUE. In this section, we further verify the effectiveness of **KnowDA** in the *Sequence labeling Tasks* which are excluded from KoMT.

**Setting** Following (Wang et al., 2022), we take **BERT-base** (Devlin et al., 2019) as the baseline model. We compare with several strong data augmentation models, including **SDANER** (Dai & Adel, 2020), **LAMBADA** (Anaby-Tavor et al., 2020), **MetaST** (Wang et al., 2021) and **PromDA** (Wang et al., 2022). We conduct the shot-10 setting with 40 samples for CoNLL'03 and 30 samples for WikiAnn. There are two stages in generating synthetic data for sequence labeling tasks: **1)** generating synthetic input sentences; **2)** labeling entities on the synthetic input sentences. We use low-resource training data to fine-tune **KnowDA** for both stages. We find that feeding the generated synthetic data back to **KnowDA** can improve its entity labeling performance and produce better synthetic data. We, thus, iterate this process times to four times. Following (Wang et al., 2022), we repeat the experiment five times with different random seeds and data splits and report the averaged results in Table 3. We also show the entity labeling performance of **KnowDA** and and the performance of BERT fine-tuned with the corresponding synthetic data in each iteration in Table 4. More details see Appendix A.4.

**Result** Table 3 illustrates BERT labeling performance on the CoNLL'03 and WikiAnn Benchmark after using synthetic data from various DA methods. Compared to the BERT baseline models, BERT with **KnowDA**'s synthetic data achieves 12.9 and 16.1 F1 score improvements. Compared to the state-of-the-art PromDA method, **KnowDA**'s synthetic data gains an improvements of 7.8 points and 8 points on CoNLL'03 and WikiAnn, respectively. Both PromDA and **KnowDA** are based on the T5-Large checkpoints. The major difference is that PromDA only receives task-agnostic pre-training on a sub-set of C4 corpus (Raffel et al., 2020), while **KnowDA** receives KoMT. This implies the effectiveness of using NLP prior task knowledge for data augmentation. As shown in Table 4, adding **KnowDA**'s synthetic data substantially improves the entity labeling performance of **KnowDA** and BERT (i.e., $T_0$ vs. $T_1$). Both models are further improved as the iteration continues. Although **KnowDA** could be directly used to perform sequence labeling tasks, the BERT model trained with generated synthetic data consistently outperforms **KnowDA** (i.e., in CoNLL'03, 85.0 vs. 85.4; in

WikiAnn, 64.9 vs. 66.3). This shows that the BERT model successfully learns the knowledge embedded in the synthetic data from **KnowDA**.

Table 3: Performance of DA on the sequence labeling benchmarks.

| Method | CoNLL'03 | Wikiann |
|---|---|---|
| BERT-base | $72.5_{4.6}$ | $50.8_{2.8}$ |
| SDANER | $73.9_{4.3}$ | $51.7_{3.3}$ |
| LAMBADA | $74.9_{3.7}$ | $52.9_{1.9}$ |
| MetaST | $76.7_{1.2}$ | $56.6_{2.3}$ |
| PromDA | $77.5_{3.5}$ | $59.0_{2.9}$ |
| **KnowDA** | $\mathbf{85.4_{0.8}}$ | $\mathbf{66.3_{2.9}}$ |

Table 4: The entity labeling performance for **KnowDA** and BERT after each iteration (from $T_0$ to $T_4$).

| Iter. | CoNLL'03 | | WikiAnn | |
|---|---|---|---|---|
| | **KnowDA** | BERT | **KnowDA** | BERT |
| $T_0$ | $80.1_{1.9}$ | $72.5_{5.1}$ | $62.5_{2.8}$ | $50.2_{3.6}$ |
| $T_1$ | $83.0_{1.3}$ | $84.0_{0.9}$ | $64.2_{3.1}$ | $65.4_{2.6}$ |
| $T_2$ | $84.9_{1.2}$ | $84.8_{0.5}$ | $65.4_{2.9}$ | $65.9_{2.8}$ |
| $T_3$ | $84.9_{0.8}$ | $85.1_{0.6}$ | $65.3_{2.9}$ | $66.0_{3.0}$ |
| $T_4$ | $85.0_{1.1}$ | $\mathbf{85.4_{0.8}}$ | $64.9_{3.0}$ | $\mathbf{66.3_{2.9}}$ |

### 3.3 DISCUSSION

In this section, we conduct an in-depth analysis of **KnowDA**. We investigate the impact of key choices and demonstrations for **KnowDA** in Sec. 3.3.1. We present a performance comparison of **KnowDA**'s fine-tune and zero-shot versions for generating long context in section Sec 3.3.2. We also compare **KnowDA** with other multi-task pre-training generative models (e.g., T0 and T5) on data augmentation in Sec 3.3.3. We further compare the synthetic data from **KnowDA** and FlipDA in Sec 3.3.4.

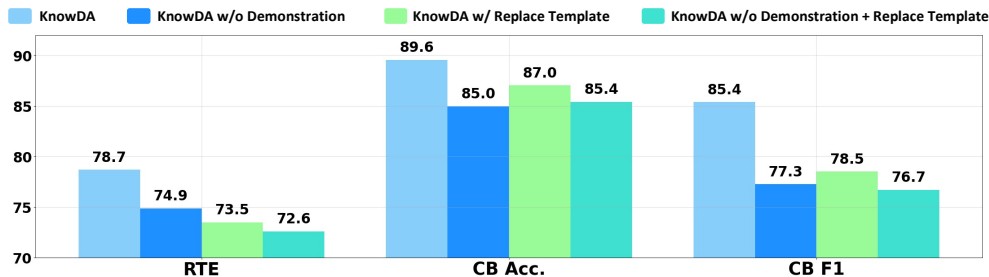

Figure 3: The impact of different choices of key and demonstrations. In *w/o Demonstration*, no demonstration examples are used. In *Replace Templates*, we use different keys to generate synthetic data. We report accuracy for the RTE task, and both accuracy and F1-score for the CB task.

### 3.3.1 TASK KNOWLEDGE TRANSFER

KoMT has two important components to transfer task knowledge: **1)** Task-specific key list; **2)** Full demonstration examples. To show the importance of key choices and demonstrations, we choose the challenging RTE and CB tasks, which involve unsupervised long document generation steps. The best template for RTE and CB long document generation is "Premise [MASK]" and "Text [MASK]", respectively. In the **Replace Template**, we replace the key *Premise* with *Context* in RTE and replace the key *Text* with *Document* in CB. We generate separated sets of synthetic data under three settings (Replace Template, w/o Demonstrations, and both). We follow the settings in Table 1 and feed these sets of synthetic data into the PET ALBERT model separately. Figure 3 shows that both key and demonstrations have a considerable impact on the generated synthetic data: RTE drops up to 6.1% accuracy, and CB F1 score drops up to 8.7%. Notably, the impact of the key is larger than the demonstrations in the RTE task, while this is opposite in the CB task. We hypothesize that this could be because keys and demonstrations activate different knowledge from **KnowDA**, which have different impacts on the downstream tasks. This is further confirmed by the fact that applying both changes would further negatively impact the PET ALBERT model performance. In summary, keys and demonstrations have similar and positive impacts to **KnowDA** in data augmentation. They effectively transfer appropriate and different prior task-specific knowledge to the ongoing task. More analysis can be found in Appendix A.7.

### 3.3.2 LONG TEXT GENERATION

As discussed in Sec. 2.4, zero-shot learning is applied when **KnowDA** generates long text for the task BoolQ, RTE, and CB. These three tasks include long text in their labelled instances (e.g., the passage in BoolQ). We conduct an ablation study and show the importance of this design choice by directly fine-tuning **KnowDA** to generate the long text. The experiment is based on the ALBERT model. As shown in Table 5, fine-tuning negatively affects the quality of synthetic data and the corresponding PET performance becomes worse than using the zero-shot version.

Table 5: Ablation Study of zero-shot generation for long text on the ALBERT-xxlarge-v2 model.

| Method | BOOLQ (Acc.) | RTE (Acc.) | CB. (Acc./F1) | avg |
|---|---|---|---|---|
| **KnowDA** (Fine-Tuning) | $73.3_{0.7}$ | $68.5_{1.8}$ | $86.3_{2.8}$ / $80.8_{6.6}$ | 77.2 |
| **KnowDA** (Original) | $78.2_{0.8}$ | $78.7_{0.9}$ | $89.6_{1.9}$ / $85.6_{2.9}$ | 83.0 |

### 3.3.3 COMPARISON WITH T0 AND RAW T5

In this section, we compare **KnowDA** with Raw T5-Large model (i.e., **KnowDA** w/o KoMT) and T0, a T5-3B model trained with a large and diverse set of human-annotated prompts to solve different NLP tasks. We select the task of WiC, WSC, and COPA for comparison because none of **KnowDA**, Raw T5 and T0 have seen these three tasks during pre-training. The experiment is based on the ALBERT model. As shown in Table 6, both raw T5 and T0 perform worse than **KnowDA**. Although T0 achieves good performance in solving those NLP tasks, T0 is never trained to produce synthetic data. When applying T0 to generate synthetic data (e.g., generating a full sentence given a class label), its data augmentation performance is sub-optimal.

Table 6: Comparison with Raw T5 and T0 model on the ALBERT-xxlarge-v2 model.

| Method | WiC | WSC | COPA | avg |
|---|---|---|---|---|
| T0 | $52.1_{2.9}$ | $71.1_{3.8}$ | $88.0_{2.8}$ | 70.4 |
| Raw T5 | $51.3_{1.1}$ | $70.2_{4.7}$ | $87.7_{2.0}$ | 69.8 |
| **KnowDA** | $55.9_{1.5}$ | $77.9_{3.6}$ | $89.0_{2.1}$ | 74.2 |

### 3.3.4 SYNTHETIC DATA QUALITY ANALYSIS

Table 1 and 2 have confirmed that the synthetic data from **KnowDA** can effectively improve low-resource performance in FewGLUE. We further examine the quality of generated synthetic data from **KnowDA** and compare that with FlipDA, including diversity analysis and human evaluation.

Table 7: Diversity analysis on Synthetic Data.

| Model | Self-BLEU↓ | Novel Entity ↑ |
|---|---|---|
| FlipDA | 67.9 | 758.5 |
| **KnowDA** | **44.0** | **1048.3** |

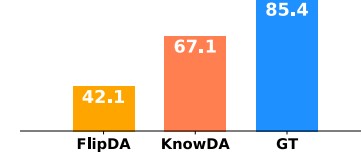

Figure 4: Human Evaluation Results.

**Diversity Analysis** We compare the diversity of the generated synthetic data from **KnowDA** and FlipDA. We sample 200 synthetic data from each FewGLUE task. Following Wang et al. (2022), we calculate **Novel Entity** (i.e., number of entity mentions or keywords not appearing in the training data) and Self-BLEU score (Zhu et al., 2018) for each FewBLUE task separately. We report the averaged score across all tasks to quantify the overall synthetic data diversity. As shown in Table 7, synthetic data from FlipDA is less diverse than the one from **KnowDA**. Its Self-BLEU score increases from 44.0 to 67.9, and Novel Entity score decreases from 1048.3 to 758.5. This explains the advantages of **KnowDA** because it provides more new training signals in the generated synthetic data.

**Human Evaluation**    We further conduct human evaluation to analyze the quality of the synthetic data. We sample 50 instances from ground-truth training data (GT), **KnowDA**'s synthetic data (**KnowDA**), and FlipDA's synthetic data (FlipDA) in each FewGLUE task, resulting in 1200 samples in total. We mask the data source of these samples and assign them to three annotators who are only told that there are slightly more machine-generated samples than human-generated ones. They are asked to judge whether each sample is generated by humans. Figure 4 shows the human evaluation results. 85% of GT are correctly classified, which shows that it is relatively easy to identify the human-generated samples. 67.1% of **KnowDA**'s synthetic data are identified as human-generated samples, while this number decreases to 42.1% for FlipDA's synthetic data. This clearly exhibits that, compared to FlipDA, the synthetic data from **KnowDA** are more similar to the human-crafted data.

## 4    RELATED WORK

**Data Augmentation for Low-resource NLP**    We divided data augmentation methods for low-resource NLP into two categories according to the complexity of the tasks that need to be augmented. The works of the first category focus on simple tasks, e.g., single-sentence classification and sequence labeling task that only has one sentence as input, and one sequence as output. Early, researchers propose word substitution based methods such as KNN replacement (Vijayaraghavan et al., 2016; Wang & Yang, 2015), Synonym Replacement (Zhang et al., 2015), TF-IDF replacement (Xie et al., 2020) and EDA (Wei & Zou, 2019b) integrated with multiple base replacements. Later, large generative models have been used for data augmentation, such as back translation (Fadaee et al., 2017; Sennrich et al., 2015) utilizes machine translation models to synthesize new data samples, LAMBADA (Anaby-Tavor et al., 2020) finetunes a GPT-2 model to generate augmented data, GPT3Mix (Yoo et al., 2021) uses GPT-3 along with hard prompting to yield augmented examples, PromDA (Wang et al., 2022) leverages soft prompting to perform efficient learning from few shots. The previous work (Zhou et al., 2021) demonstrates that these methods are hard to generate proper synthetic data for tasks with more complicated structures (e.g., SuperGLUE). Another line of low-resource data augmentation is exploring augmentating tasks with complicated structures (i.e., long sequences or multiple sentences). FlipDA (Zhou et al., 2021) has conducted preliminary attempts on these tasks by applying local phrase-level edits over the original training data. Our method also make efforts on this setting. The main difference between **KnowDA** and FlipDA lies in that our **KnowDA** could generate the whole augmented examples *from scratch*.

**Multi-task Pre-training**    Recent research efforts have emerged in unifying multiple NLP tasks as a unified Sequence-to-Sequence text generation task. Paolini et al. (2021) proposes to tackle various structured prediction language tasks using a single text-to-text generation model. Wei et al. (2021), Sanh et al. (2022), Aribandi et al. (2022), and Xu et al. (2022) further demonstrate that adding these unified Sequence-to-Sequence NLP training instances to pre-trained language models can further improve the model performance. Our model is broadly inspired by this long line of previous works with following major differences. First, we focus on utilizing more human-annotated NLP task datasets to learn how to generate synthetic samples of a given task rather than just learning how to complete the task. Second, most of previous works rely on the human-crafted task-specific hard prompt as the unified format to do a multi-task pre-training. Our unified key-value format does not require extra human efforts and allows KoMT to easily scale to many NLP tasks.

## 5    CONCLUSION AND FUTURE WORK

This paper explores multi-task learning paradigms at a massive scale for data augmentation in low-resource NLP tasks for the first time. We demonstrate that the proposed Knowledge Mixture training enables pre-trained language models the capability of generating proper synthetic instances *from scratch* for complicated tasks (i.e., the data sample has long sequences or multiple sentences). Experiments verified the effectiveness of our **KnowDA**, and **KnowDA** outperforms state-of-the-art data augmentation approaches on the popular benchmarks FewGLUE, CoNLL'03, and WikiAnn. We also perform ablation studies indicating the importance of including demonstrations and the impact of different keys. Moreover, increasing the size of multi-task scaling and investigating more advanced training objectives for data augmentation is still a promising direction worthy of long-term exploration.

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

# A   APPENDIX

In this Appendix, we provide more experiment details about **KnowDA** in Sec. A.1. We introduce the diverse NLP tasks used in KoMT in Sec. A.3. We add additional experiment description about our data augmentation experiments in Sec. A.2 and A.4. We present the detailed data augmentation procedure in Sec. A.5. Finally, in Sec. A.6, we showcase representative FewGLUE examples generated by **KnowDA**.

## A.1   EXPERIMENT DETAILS

**Demonstration Selection in KoMT**   As we discussed above, we use demonstration in KoMT to allow **KnowDA** to transfer task-specific knowledge. Given the training NLP instance $T$, we first conduct random masking to it and treat the resulting string as initial input string. We then select 16 instances from the same NLP task and check to see the maximum number of instances (i.e., $m$) can be inserted into the input string within the maximum input length. Lastly, we will randomly select $k$ instances as demonstration where $k$ is also a random number between 0 and $m$, which avoids **KnowDA** from relying on the demonstration too much. The only requirement for the demonstrations is that they should come from the same NLP task as the masked NLP task instance. In addition, even in KoMT, long NLP instance (i.e., input length close to maximum input length) may never be accompanied by any full demonstration examples.

**Soft Prompt for Data Augmentation**   We observe that, when **KnowDA** is trained to generate input text features, **KnowDA** generates relatively long text (e.g., a document for QA); when **KnowDA** is trained to generate output labels, it normally generates relatively short text (e.g., label phrases). To *disentangle* the different skills required for different roles, during KoMT, we prepend two sets of *Soft Prompt* (i.e, randomly-initialized trainable vectors) at the beginning of each layer. One for input text generation and another one for output text generation. Different from the standard prompt tuning, we train the parameters of **KnowDA** and these *Soft Prompt* together. We add 5 vectors per layer, resulting 5 * 24 * 1024 * 2 = 245,760 additional parameters (only 0.003% of the original T5-Large model). When applying **KnowDA** to downstream tasks, we should use the appropriate *Soft Prompt* for unsupervised generation or fine-tuning. Our preliminary experiments show that using different *Soft Prompt* has an significant impact on the overall performance.

## A.2   FEWGLUE

In this section, we add more implementation details for the FewGLUE data augmentation experiments in Sec. 3.1. Appendix A.5 shows the details of data augmentation procedure for each FewGLUE task. In all FewGLUE tasks, when we need to update the parameters of **KnowDA**, we simply fine-tune **KnowDA** (i.e., updating all parameters) with batch size 12 with learning rate of $5e^{-6}$ for 500 steps. For fair comparison with FlipDA, we directly produce synthetic data and feed into the ALBERT and DeBERTa model for further training. Unlike FlipDA which creates synthetic instances from the few-shot instances, **KnowDA** produces synthetic instances *from scratch*. Consequently, there is no explicit linkage between the synthetic instances and the few-shot instances. We therefore use the ALBERT and DeBERTa model for *Consistency filtering*, which only keeps synthetic instances that have the consistent output results from **KnowDA** and the ALBERT and DeBERTa model. We find such filtering policy works as well as the complicated label-based filtering strategies proposed in FlipDA Zhou et al. (2021).

In this experiment, we use PET (either based on the ALBERT or DeBERTa model) as the base model. PET is a semi-supervised training algorithm that uses language-based patterns to convert the input training instances into cloze-style. For example, to predict whether "Mia likes pie" and "Mia hates pie" are entailed or not, PET covert them into "Mia likes pie __ Mia hates pie". The training objective is to train the PET model to fill "Yes" or "No". Note that given a target task, there could be multiple patterns. PET works as follows: 1) we train a separate PET model for each pattern on the same small training data; 2) The ensemble of all the above models is then combined to annotate a large unlabeled dataset; 3) the labeled dataset above is finally used to train a final classifier.

## A.3  KoMT Task Details

Table 8 and 9 list all supervision NLP tasks in KoMT. They are all available from Huggingface Dataset Lhoest et al. (2021). Popular task types include QA, Multiple Choices, Text Classification, Text Summarization and Generation and Machine Reading comprehensive.

Table 8: All of the training NLP task datasets used in KoMT.

| Dataset(s) | Description | No. Train Datasets | $|\mathcal{D}|$ | Citation |
|---|---|---|---|---|
| ADECorpusV2 | Text Classification | 3 | 23,516 | Gurulingappa et al. (2012) |
| Adversarial-QA | Natural Language QA | 1 | 30,000 | Bartolo et al. (2020b) |
| AESLC | Email Summarization | 1 | 14,436 | Zhang & Tetreault (2019) |
| AG-News | Text Classification | 1 | 120,000 | Zhang et al. (2015) |
| AI2-ARC | Multiple Choice | 1 | 1,119 | Clark et al. (2018) |
| ANLI | Adverserial NLI | 1 | 162,865 | Nie et al. (2020) |
| AppReviews | Text Scoring | 1 | 288,065 | Grano et al. (2017) |
| Aqua-RAT | Multiple Choice | 1 | 97,467 | Ling et al. (2017) |
| ART | NLI | 1 | 169,654 | Bhagavatula et al. (2019) |
| ASLG-PC12 | Text Generation | 1 | 87,710 | Othman & Jemni (2012) |
| BioMRC | Machine Reading Comprehension | 1 | 700,000 | Pappas et al. (2020) |
| Break-Data | Question Decomposition Meaning Representations | 2 | 61,824 | Wolfson et al. (2020) |
| Circa | Text Classification | 1 | 34,268 | Louis et al. (2020) |
| Climate-Fever | Text Scoring | 1 | 1,535 | Diggelmann et al. (2020) |
| Codah | Multiple Choice | 1 | 1,665 | |
| Common-Gen | Text Generation | 1 | 67,389 | Lin et al. (2019a) |
| Commonsense-QA | Open Domain QA | 1 | 9,741 | Talmor et al. (2018) |
| COS-E | Text Generation | 1 | 9,741 | Rajani et al. (2019) |
| Cosmos-QA | Multiple Choice | 1 | 25,262 | Huang et al. (2019) |
| DBpedia14 | Text Classification | 1 | 560,000 | Lehmann et al. (2015) |
| Definite-Pronoun-Resolution | Pronoun resolution | 1 | 1,322 | Rahman & Ng (2012) |
| Discovery | Text Classification | 1 | 1,566,000 | Sileo et al. (2019) |
| Dream | Multiple Choice | 1 | 6,116 | |
| DuoRC | Abstractive QA | 1 | 60,721 | Saha et al. (2018) |
| E2E-NLG-Cleaned | Text Generation | 1 | 33,525 | Dušek et al. (2020) |
| ELI5 | Abstractive QA | 1 | 502,937 | Fan et al. (2019) |
| EMO | Text Classification | 1 | 30,160 | Chatterjee et al. (2019) |
| Emotion | Text Classification | 1 | 16,000 | Saravia et al. (2018) |
| Empathetic-Dialogues | Open Domain QA | 1 | 76,673 | Rashkin et al. (2019) |
| Financial-Phrasebank | Text Classification | 1 | 2264 | Malo et al. (2014) |
| FreebaseQA | Open Domain QA | 1 | 20,358 | Jiang et al. (2019) |
| Gigaword | Text Summarization | 1 | 3,803,957 | Rush et al. (2015) |
| GLUE | General Language Understanding | 7 | 949,101 | Wang et al. (2019) |
| GoogleWellformedQuery | Text Scoring | 1 | 17,500 | Faruqui & Das (2018) |
| HateSpeech18 | Text Classification | 1 | 10,944 | De Gibert et al. (2018) |
| HateSpeechOffensive | Text Classification | 1 | 24,783 | Davidson et al. (2017) |
| Hatexplain | Text Classification | 1 | 15,383 | Mathew et al. (2020) |
| HealthFact | Text Classification | 1 | 9,832 | Kotonya & Toni (2020) |
| Hellaswag | Text Generation | 1 | 39,905 | Zellers et al. (2019) |
| HotpotQA | QA | 1 | 90,447 | Yang et al. (2018) |
| IMDb Reviews | Text Classification | 1 | 25,000 | Maas et al. (2011) |
| Jeopardy | Natural language QA | 1 | 216,930 | |
| KILT | Knowledge-Intensive Language Tasks | 6 | 2,731,679 | Petroni et al. (2021) |
| Liar | Fake News Detection, Text Classification | 1 | 10,269 | Wang (2017) |
| Limit | Text classification | 1 | 23,559 | Manotas et al. (2020) |
| MathQA | QA | 1 | 29,837 | Amini et al. (2019) |
| MC-Taco | Multiple Choice | 1 | 9,442 | Zhou et al. (2019) |
| Medical-Questions-Pairs | Text Classification | 1 | 3048 | McCreery et al. (2020) |
| Mocha | QA | 1 | 31,069 | Chen et al. (2020) |
| Multi-News | Text Summarization | 1 | 44,972 | Fabbri et al. (2019) |
| NumerSense | Numerical commonsense reasoning probing | 1 | 10,444 | Lin et al. (2020) |
| Onestop-English | Text Classification | 1 | 567 | Vajjala & Lučić (2018) |
| OpenBookQA | Open Domain QA | 1 | 4,957 | Mihaylov et al. (2018) |
| PAWS | Text Classification | 1 | 49,401 | Zhang et al. (2019) |
| PIQA | Multiple Choice | 1 | 16,000 | Bisk et al. (2020) |
| Poem-Sentiment | Text Classification | 1 | 892 | Sheng & Uthus (2020) |
| QA-SRL | Multiple Choice | 1 | 6,414 | He et al. (2015) |
| QASC | Multiple Choice | 1 | 8,134 | Khot et al. (2020) |
| QUAIL | Multiple Choice | 1 | 10,246 | Rogers et al. (2020) |
| QUAREL | Multiple Choice | 1 | 1,941 | Tafjord et al. (2019a) |
| QUARTZ | Multiple Choice | 1 | 2,696 | Tafjord et al. (2019b) |
| QUOREF | QA | 1 | 19,399 | Dasigi et al. (2019) |
| RACE | School QA (MCQ) | 2 | 87,866 | Lai et al. (2017) |
| Reddit-Tifu | Text Summarization | 1 | 42,139 | Kim et al. (2018) |
| Ropes | Extractive-QA | 1 | 10,924 | Lin et al. (2019b) |
| Rotten-Tomatoes | Text Classification | 1 | 8,530 | Pang & Lee (2005) |
| SamSum | Text Summarization | 1 | 14,372 | Gliwa et al. (2019) |
| Sentiment140 | Text Classification | 1 | 1,600,000 | Go et al. (2009) |
| SCICite | Text Classification | 1 | 8,194 | Cohan et al. (2019) |
| SCIQ | Multiple Choice | 1 | 11,679 | Welbl et al. (2017) |
| SCITail | NLI | 1 | 23,596 | Khot et al. (2018) |
| Search-QA | QA | 1 | 151,295 | Dunn et al. (2017) |
| Sick | Text Classification | 1 | 4,439 | Marelli et al. (2014) |

Table 9: All of the training datasets used in KoMT Part II.

| Dataset(s) | Description | No. Train Datasets | $|\mathcal{D}|$ | Citation |
|---|---|---|---|---|
| SMS-Spam | Text Classification | 1 | 5,574 | Almeida et al. (2011) |
| Social-I-QA | QA | 1 | 33,410 | Sap et al. (2019) |
| Spider | Text Generation | 1 | 7,000 | Yu et al. (2018) |
| SQuAD | QA (context) | 1 | 87,599 | Rajpurkar et al. (2016) |
| Swag | Text Classification | 1 | 73,546 | Zellers et al. (2018) |
| Tab-Fact | Text Classification | 1 | 92,283 | Chen et al. (2019) |
| TREC | Text Classification | 1 | 5,452 | Li & Roth (2002) |
| TweetEval | Text Classification | 9 | 120,104 | Mohammad et al. (2016) |
| TweetQA | Natural Language QA | 1 | 10,692 | Xiong et al. (2019) |
| WebQuestions | QA (open) | 1 | 3,778 | Berant et al. (2013) |
| WIQA | QA | 1 | 29,808 | Tandon et al. (2019) |
| Wiki-Bio | Text Generation | 1 | 582,659 | Lebret et al. (2016) |
| Wiki-QA | QA | 1 | 20,360 | Yang et al. (2015) |
| Wiki-Split | Text Split | 1 | 989,944 | Botha et al. (2018) |
| WikiSQL | NLI | 1 | 56,355 | Zhong et al. (2017) |
| XSum | Text Summarization | 1 | 203,577 | Narayan et al. (2018) |
| Yelp-Polarity | Text Classification | 1 | 560,000 | Zhang et al. (2015) |
| YelpReviewFull | Text Classification | 1 | 650,000 | Zhang et al. (2015) |
| Total | | 114 | | - |

## A.4 SEQUENCE LABELING TASKS

In this section, we add more implementation details for the sequence labeling data augmentation experiments in Sec. 3.3, including data augmentation procedure, training technologies and iteration-based training.

**Data Augmentation Procedure** `KnowDA` produces synthetic data via two steps:

1. Generating input sentences (without any entity annotations) via the sequence of named entity tags. This is similar to the *Output View* described in Wang et al. (2022).

2. Use `KnowDA` to label named entities for the generated input sentences.

For example, given a training sequence labeling data instance:

[*Org* **All Fishermen 's Association**] secretary [*Per* **N.J. Bose**] said the strike would continue indefinitely.

where ***Org*** corresponds to the entity label ***Organization*** and ***Per*** corresponds to the entity label ***Person***. In the **Stage 1**, the format of a training instances is:

Input: [Output Tags]: Organization and Person [Sentence]: ⟨MASK⟩
Target: All Fishermen 's Association secretary N.J. Bose said the strike would continue indefinitely.

As sequence labeling tasks mostly involves with short sentences, we train a copy of `KnowDA` to complete this step. In the **Stage 2**, following Aribandi et al. (2022), the format of a training instances is:

Input: [Output Tags]: ⟨MASK⟩ [Sentence]: All Fishermen 's Association secretary N.J. Bose said the strike would continue indefinitely.
Target: Organization All Fishermen 's Association; Person N.J. Bose.

Given the target outputs, we can easily map the entity strings back to the input sentences and obtain word-level BIO tags. Although this approach may fail when entity appears multiple times in the input sentence, we find such case rarely impact on the final labeling performance. We train a separated copy of `KnowDA` to complete this step.

**Training KnowDA for Data Augmentation**    Interestingly, similar to Wang et al. (2022), we find it beneficial to add *Soft Prompt* (i.e., trainable vectors prepended to each layer of PLM) when training **KnowDA** to label entities. In this paper, we directly use randomly-initialized *Soft Prompt* (rather than the pre-trained *Soft Prompt* in PromDA). When we train **KnowDA** to label entities in Stage 2, we frozen all parameters of **KnowDA** and only train the newly added *Soft Prompt* parameters with a learning rate of $1e^{-3}$. When we train **KnowDA** to generate input sentences, we simply fine-tune all **KnowDA** parameters with a learning rate of $5e^{-6}$.

**Iteration-based Training**    Given the low-resource training data, we first train **KnowDA** to generate synthetic input sentences without entity annotations, then train another copy of **KnowDA** to label entities over the synthetic input sentences. Interestingly, feeding these complete synthetic data, combined with few-shot training data, back to **KnowDA** can improve its entity labeling performance. **KnowDA** with stronger entity labeling performance can be used to annotate the synthetic input sentences again and produce better synthetic data. We thus iterate this process four times. In every iteration, the synthetic input sentences are fixed and we only update the entity annotations on these sentences. The iteration results, including **KnowDA**'s entity labeling performance and the performance of the BERT model trained with the corresponding synthetic data are shown in Table 4. These results are all averaged across five different random seeds and data splits.

## A.5 FewGLUE Data Augmentation Procedure

In the task of BoolQ, RTE, CB, MultiRC and ReCoRD, we use **KnowDA** to generate long text without any further fine-tuning (i.e., zero-shot). All of these zero-shot components are in Blue with the **snow** mark. All components with fine-tuning are in Orange with the **fire** mark.

**BoolQ**    As shown in Figure 5, the data augmentation procedure for the task of BoolQ has three steps: i) generating article; ii) generating question and iii) generating the final answer (i.e., True or False). As **article** in BoolQ are often relatively long, we generate it directly from **KnowDA** without any fine-tuning (e.g., zero-shot). We further separately fine-tune **KnowDA** for the rest two steps.

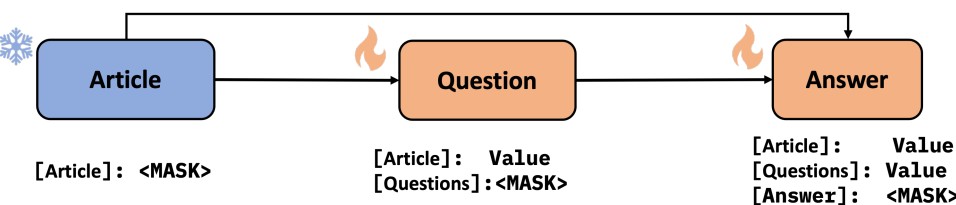

Figure 5: The Data Augmentation Procedure for the task of BoolQ.

**RTE**    As shown in Figure 6, the data augmentation procedure for the task of RTE has three steps: i) generating premise; ii) generating hypothesis and iii) generating the final relation label (i.e., whether premise and hypothesis are entailment). As **premise** in RTE are often relatively long, we generate it directly from **KnowDA** without any fine-tuning (e.g., zero-shot). We further separately fine-tune **KnowDA** for the rest two steps.

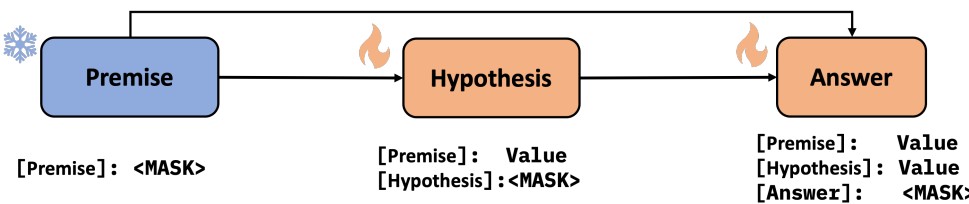

Figure 6: The Data Augmentation Procedure for the task of RTE.

**CB**  As shown in Figure 7, the data augmentation procedure for the task of CB has three steps: i) generating premise; ii) generating hypothesis and iii) generating the final relation label (i.e., whether premise and hypothesis are entailment). As **premise** in CB are often relatively long, we generate it directly from **KnowDA** without any fine-tuning (e.g., zero-shot). The premise text in CB are often dialogue-based, which is different from commonly seen premise in other NLI tasks. We therefore use the key "text" and full demonstrations to generate synthetic data for CB premise. We further separately fine-tune **KnowDA** for the rest two steps.

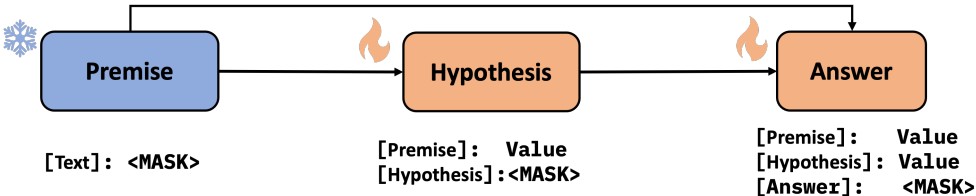

Figure 7: The Data Augmentation Procedure for the task of CB.

**MultiRC**  As shown in Figure 8, the data augmentation procedure for the task of MultiRC has four steps: i) generating document; ii) generating question; iii) generating answer and iv) generating label (i.e., whether the article, question and answer are correct). As **document** in MultiRC are often relatively long, we generate it directly from **KnowDA** without any fine-tuning (e.g., zero-shot). We further separately fine-tune **KnowDA** for the rest three steps.

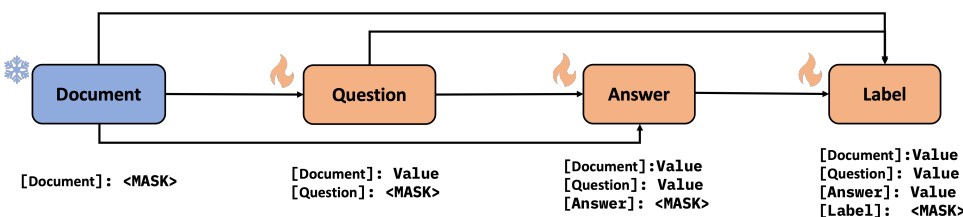

Figure 8: The Data Augmentation Procedure for the task of MultiRC.

**WiC**  As shown in Figure 9, the data augmentation procedure for the task of WiC has two steps: i) generating the disambiguation word as well as the sentence pair containing the above word for selection; ii) generating final result (i.e., whether the words in the sentence pair have the same meaning). We fine-tune **KnowDA** for both steps.

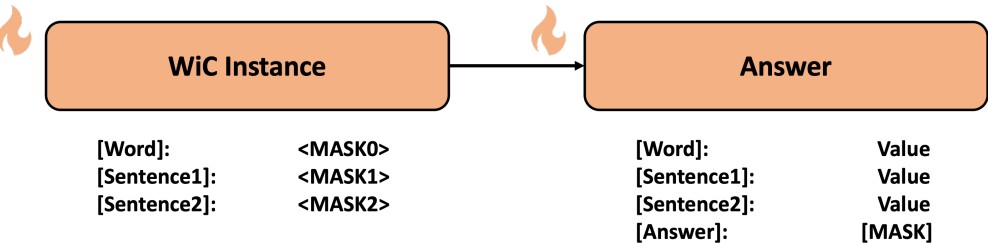

Figure 9: The Data Augmentation Procedure for the task of WiC.

**WSC**  As shown in Figure 10, the data augmentation procedure for the task of WSC has two steps: i) generating the sentences that include the noun (e.g., person name) and pronoun which refers to the previous noun; ii) generating final result (i.e., whether the noun and pronoun refer to the same entity, such as specific person). We fine-tune **KnowDA** for both steps.

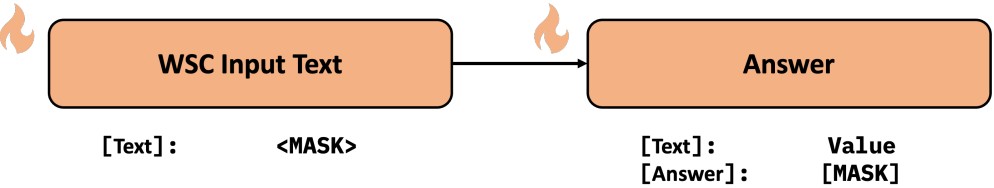

Figure 10: The Data Augmentation Procedure for the task of WSC.

**COPA**   As shown in Figure 11, the data augmentation procedure for the task of COPA has two steps: i) generating new premise text given the options and questions. This is because given the premise and question, there could be many possible options to be generated. We find those generated synthetic data contributes little to the ALBERT and DeBERTa model performance in our preliminary experiments; ii) generating final answer (i.e., which option is correct given the premise and question). We fine-tune **KnowDA** for both steps.

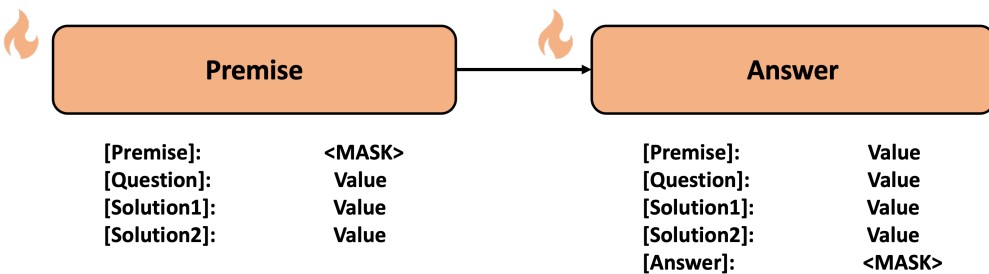

Figure 11: The Data Augmentation Procedure for the task of COPA.

**ReCoRD**   As shown in Figure 12, the data augmentation procedure for the task of ReCoRD has four steps: i) generating document; ii) generating entity list; iii) generating query text and iv) verifying the answer by generating the entity. As **document** in ReCoRD are often relatively long, we generate it directly from **KnowDA** without any fine-tuning (e.g., zero-shot). We further separately fine-tune **KnowDA** for the rest three steps. When generating query, we generating full sentences without any placeholder. We will create placeholder for the query text if there is an entity appearing in the query text and the entity list.

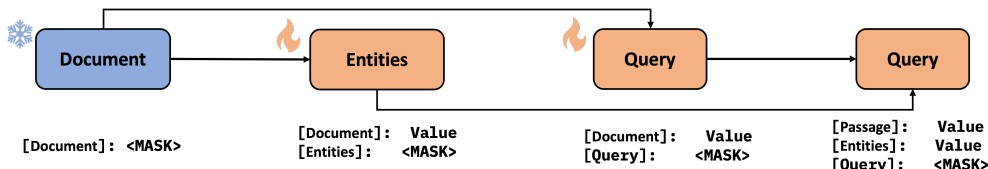

Figure 12: The Data Augmentation Procedure for the task of ReCoRD.

A.6   CASE STUDY

In this section, we select eight representative examples from **KnowDA** in the FewGLUE benchmark (i.e., one example for each task). In the task of BoolQ, RTE, CB, MultiRC and ReCoRD, we use **KnowDA** to generate long text without any further fine-tuning (i.e., zero-shot). We show that **KnowDA** could produce high-quality text with similar styles in this challenging setting. In the task of COPA, WSC and WiC, we find that after task-specfic fine-tuning, **KnowDA** could also generate appropriate text.

**BoolQ** In the original BoolQ training data, there are some instances about biography of famous athletes. **KnowDA** successfully produce another detailed, coherent and long athlete biography from the keys and demonstration examples, showing the success of KoMT.

> **Article**: Selena Grace Anderson (born December 9, 1987) is an American football wide receiver for the University of Alabama at Birmingham (UAB). She was drafted by the New Orleans Saints in the third round of the 2011 NFL Draft. She played college football at Syracuse. **Questions**: Is Selena Grace a basketball player? **Answer**: False

**RTE** Similar to the BoolQ task, many instances in the RTE task have long value in **Premise**. **KnowDA** successfully produce long and high-quality text for Premise only using key and demonstrations.

> **Hypothesis**: Fort Lauderdale is part of the United States. **Premise**: The Millennium State Highway (the New Millennium) is a west-bound U.S. highway that runs for 183 miles (195 km) between the cities of Fort Lauderdale and Fort Lauderdale-Davis, Florida. Developed by the Federal Highway Administration in 1956, it is one of the major arterial roads in the U.S. **Answer**: entailment

**CB** In the CB task, unlike above Wikipedia-style text, some instances are drawn from special domains, such as fiction (i.e., from British National Corpus) and spoken dialogue (i.e., from Switchboard). **KnowDA** still generate long text following these special styles.

> **Hypothesis**: Jools Holland was the director of the movies. **Premise**: And who is the director of the movies in England? said Jools Holland who said he hoped we'd get some film or television. **Answer**: contradiction

**COPA** In this example, the original premise in the training is "The customer filed a complaint with the store manager.". Given options and question, **KnowDA** could generate different but coherent premise text.

> **Premise**: The customer was upset. **Solution1**: The sales associate undercharged the customer. **Solution2**: The sales associate acted rude to the customer. **Question**: What is the cause for this? **Answer**: 2

**WSC** In the WSC task, **KnowDA** directly generates the whole Premise text with the noun and pronoun work marked with "*". The generated text are coherent: the pronoun "he" clearly refers to "Toby". This shows that **KnowDA** has strong few-shot ability because it could quickly learn this pattern from only 32 examples.

> **Premise**: Stan's son *Toby* was playing football with his friends when *he* was injured . **Answer**: True

**WiC** In the WiC task, **KnowDA** directly generates word and the two sentences. The generated results follow the WiC data format: the sentence pair needs to include the word *hand*. This shows that **KnowDA** has strong few-shot ability because it could quickly learn this pattern from only 32 examples.

> **sentence1**: You look like someone hand you a book. **sentence2**: I pushed a hand against a table. **word**: hand **Answer**: False

**MultiRC** **KnowDA** further demonstrate its ability in generating long text. The style of the article is similar to the training data in MultiRC. **KnowDA** also generates high-quality questions given the long article. This shows that **KnowDA** could make use the same skills for different tasks (e.g., in BoolQ, **KnowDA** also generates good questions).

> **Article**: The women and two men were abducted on Thursday by students on their way to class in the Cambridge St Andrews Catholic Primary School. The child had been released in April after getting school clothes, but police believe there was too much blood on her

leg. The condition of the pupils is not known. The Catholic school, which opened in 1962, has a special focus on education. Sisters and brothers from the school remain trapped in the school, a service which has been called a miracle. It is the first such ever celebration of modern education. **Question**: What school opened in 1962? **Answer**: Cambridge St Andrews Catholic Primary School **Label**: True

**ReCoRD**    In the ReCoRD task, there are a few instances talk about Ukraine. Interestingly, through the demonstration, **KnowDA** successfully generates another high quality piece of text regarding this topic.

> **Passage** Ukraine's prime minister has said it is not encouraging tense talks between President Vladimir Putin and the opposition in order to expel the dissident republican leader. Ukraine's chief prosecutor accused Mr Putin of refusing to cooperate with the independent opposition, alleging that he had been building a grudge against the republican group. He was confronted in Chechnya on Sunday, but was not allowed to hold talks with the opposition. Ukraine was moved to emergency status as a result of its violations of the ceasefire with Russia after protests in Kiev and Shenyang over the halt to other steps. President Vladimir Putin denies intelligence evidence that Kiev has been preparing a terror campaign to combat the rebel group. Ukraine said in the statement on Monday that it was preparing a prosecution, though there is no evidence linking the attacks to what is known as a parallel deal between the two main parties in the region. The two sides' treatment of each other in Kiev is said to have been a hostile act, with officers accusing Putin of triggering a war, while the opposition, which is not a ally of the Russians, denies any involvement. **Query** Mr Putin is not allowed to hold talks with the opposition in @placeholder because it would violate the ceasefire, Yuriy Yatsenyuk said. **Answer** Chechnya

## A.7    THE IMPACT OF THE NUMBER OF DEMONSTRATION EXEMPLARS

In this experiment, we expand the analysis in Section 3.3.1 to investigate the impact of the number of demonstration exemplars. Table 10 shows the PET ALBERT model performance in the task of RTE and CB when using $k$ demonstration exemplars where $k = 0,1,2,3$. The performance gap between the **KnowDA** model without exemplars and the one with exemplars is much larger than the **KnowDA** models using a different number of exemplars. Thus, it is important to use exemplars in the **KnowDA** model, but its data augmentation performance is relatively robust to the choices of the number of exemplars.

Table 10: The impact of the number of demonstration exemplars in the PET ALBERT model.

| Method | RTE (Acc.) | CB. (Acc./F1) |
|---|---|---|
| 0 exemplar | 74.9 | 85.0 / 77.3 |
| 1 exemplar | 78.1 | 87.7 / 83.9 |
| 2 exemplars | 78.7 | 89.6 / 85.4 |
| 3 exemplars | 78.9 | 89.1 / 85.0 |

