# OpenReview forum: "KnowDA: All-in-One Knowledge Mixture Model for Data Augmentation in Low-Resource NLP"
_ICLR.cc/2023/Conference — ICLR 2023 poster_

### Official Review · Reviewer_Fev8 · 2022-10-21

**Confidence:** 3
**Correctness:** 4
**Technical Novelty And Significance:** 3
**Empirical Novelty And Significance:** 3
**Recommendation:** 8

**Clarity, Quality, Novelty And Reproducibility:**

Even though the manuscript requires some deep understanding of the specific problem of low-resource NLP tasks and some previous methods, such as FlipDA and PET, if we consider the page limitation, the authors do a great job of trying to make the core ideas clear and understandable. I would have appreciated some extra details on some previous concepts that are key for the understanding, such as how PET handles semi-supervised training, but I understand the space is limited, and any other reader well-versed on this concrete field will perfectly understand. Beyond that, the manuscript's quality is high, it is really well written, and all provided figures and tables are clear and help further understand all the ideas. And for the novelty, despite not having a deep understanding of previous approaches, I believe the authors advance sufficiently the state of the art, and present enough, and well-justified innovations, to claim that they are making a contribution to the field.

I cannot assess the reproducibility, but I guess the authors will be able to provide at least the core model after KoMT training as a pretrained LM model, which will help future researchers reproducing all the results. Still, it is to question what happens with the fine-tuned models, used for autoregressive generation. Here actually is where I have some questions to the authors. You train a different copy of KnowDA for generation, at each stage, for each NLP task. This seems extremely laborious and time and space-consuming. Therefore:- Could you elaborate more on this, and present more precise numbers on how many fine-tuned models are then obtained?
- How much time does it require to perform these fine tunings? Are those performed on the fly, with the generation, and then discarded?
- In case these models are stored, are they saved as the diff on the weights with respect to the parameters of the "base"-KoMT KnowDA?

Besides, I would have another question, just out of curiosity:
- In the case of long text generation, should not fine-tuning plus transferring task knowledge be even more efficient? If not, why?

**Strength And Weaknesses:**

The paper presents interesting and meaningful methodologies for the important problem of tackling low-resource NLP tasks. All the ideas proposed, even though based on previous well-known works, still offer a substantial degree of advancement, and move in reasonable and intuitive directions, which is always appreciated when discussing large LMs. The core idea of KnowDA, of harmonizing tasks' representation and training objectives, is clear, well-argued and nicely implemented. Besides, some additional tricks, such as the zero-shot plus demonstrators for long text generation, and the autoregressive approach for better generation, are quite reasonable and grounded to tackle some of the most complicated and challenging tasks.

Furthermore, the authors do an excellent job in terms of exhaustively benchmarking their methodology, both from the point of view of the specific tasks considered, varied and challenging, and the number of baselines they compare against. And in most of these tasks, the methods proposed outperform previous ones. And when it is not the case, the authors try to give some intuition on why the methodology might down-perform in these cases, which is appreciated. From my point of view, the results are quite good, and prove the proposed methodology advances in the right direction in the problem of low-resource NLP.

Finally, I also appreciate the extensive work done for the supplementary material, as it helps clarify many of the doubts that might arise during the main manuscript, and provide further valuable results and information to better understand the value of the current method.


**Summary Of The Paper:**

In the current paper, the authors present a new framework to perform data augmentation on low-resource NLP tasks. The core innovation is the idea of Knowledge mixture training (KoMT), that allows representing a large variety of tasks in a uniform way, in the form of keys and values pairs. This allows training the LM in a large range of NLP tasks through a denoising objective, where different combinations of full keys or values are masked. Through additional tools, such as the usage of demonstrators, and by also harnessing the zero-shot capabilities of the large LM model used as initial checkpoint (T5), the authors show the capabilities of KnowDA on tasks that require generating long text sequences. Furthermore, at generation time, the authors propose an autoregressive scheme, through some fine-tuning of the model after KoMT, to better adapt to the structure of each task, by establishing a dependency relation between the different features of the task at hand.

Through all the proposed innovations, the authors are able to provide significant improvement on many low-resource NLP tasks, more notably the fewGLUE benchmark, when compared to previous approaches such as FlipDA. Besides, they perform several ablation studies to underline the significance of the proposed methods.

**Summary Of The Review:**

I believe the authors confront a challenging problem, not from the point of view of a reduced set of tasks, but from a really holistic perspective, tackling a large set of challenging NLP problems. The results reported are quite competitive, outperforming substantially previous approaches. Besides, the paper is really well written, and provides a clear intuition of the advancements implemented, understandable also for the readers non-versed on this particular field of NLP. For these reasons, I believe the paper paper is quite relevant, and should be accepted as it is.

---

### Official Review · Reviewer_CnYA · 2022-10-24

**Confidence:** 4
**Correctness:** 3
**Technical Novelty And Significance:** 3
**Empirical Novelty And Significance:** 3
**Recommendation:** 6

**Clarity, Quality, Novelty And Reproducibility:**

The paper provides an appendix to show more details for method and experiment designs. Code is also included to facilitate reproducibility.


**Strength And Weaknesses:**

Strength:

+The key-value list format to unify different NLP tasks for multi-task learning is interesting and seems helpful.

+KnowDA achieves strong performance on the benchmark datasets for low-resource learning.

Weakness:

-Although interesting, the key-value list format is similar to the text generation formulation of NLP tasks that is studied extensively recently (e.g., [1]). These work should be discussed to highlight the model's benefits.

-Compared to the baselines (e.g., FlipDA) that do not require training data of external NLP tasks, KnowDA needs to use datasets of multiple NLP tasks to pre-train its model. This might make the comparison less compatible and convincing in this work.

-Given the training data of multiple NLP tasks for pre-training KnowDA, another possible baseline is to formulate the NLP tasks into the text-to-text problems using the key-value list format. Given a target NLP task, a text-to-text model can be trained over the external task data and the provided training data to directly solve the task (i.e., not using data augmentation). It might be helpful to discuss or evaluate this baseline to show the benefits of data augmentation given training data from external tasks.

[1] Paolini et al. Structured prediction as translation between augmented natural language, 2021.

**Summary Of The Paper:**

This paper explores a data augmentation method, called KnowDA, for low-source NLP tasks, aiming to capture task knowledge to improve the relevance and diversity of augmented data. The method first attempts to pre-train text-to-text generation model that will be used to generate augmented data for NLP tasks for data augmentation. To learn from diverse NLP tasks, the method select multiple NLP tasks with public datasets in Huggingface. Afterward, a key-value list format is introduced as a unified framework to represent instances of different NLP tasks. The key idea is to use keys to indicate feature functions (e.g., premise, hypothesis, tag) and values to capture string representations of feature content. KnowDA then uses denoising objectives to pre-train the model over converted instances of the NLP tasks (i.e., via masking selected values in the key-value list format for reconstruction from remaining information). In addition, KnowDA adds some selected demonstrations/instances from the same tasks to the input for the text-to-text formulation during pre-training. To leverage KnowDA for data generation, task-specific feature dependency is used to provide an order to generate keys and values to create new training data for a target task. To generate long values (e.g., documents), KnowDA uses zero-shot learning framework. The method is evaluated over low-resource experiments using FewGLUE, CoNLL'03 and WikiAnn (for sequence labeling). Experiments show better performance of KnowDA over several selected baselines. The paper also conduct some analysis to demonstrate the benefits of KnowDA, including ablation study, long text generation, data diversity, and human evaluation.


**Summary Of The Review:**

The fairness of the comparison can be improved as KnowDA uses training data from multiple external NLP tasks that is not employed in the baselines. A text-to-text baseline can be considered to show the benefits of data augmentation.

---

### Official Review · Reviewer_U7D8 · 2022-10-25

**Confidence:** 4
**Correctness:** 3
**Technical Novelty And Significance:** 3
**Empirical Novelty And Significance:** 3
**Recommendation:** 6

**Clarity, Quality, Novelty And Reproducibility:**

The paper is well written.
The proposed KoMT pre-finetuning method, and KnowDA model is novel.
The experiments should be mostly reproducible.

**Strength And Weaknesses:**

Strengths:

The paper is well written and easy to follow;
The proposed KoMT pre-finetuning technique is novel and the paper claims to be the first work to perform large scale multi-task pre-training on over 100+ NLP tasks;
The model is thoroughly evaluated, and shows improved data augmentation performance on low-resource NLP tasks. Human evaluation demonstrates that the generated data samples are closer to human generated samples than previous SOTA models.

Weaknesses:

1. Despite the improved data generation performance, several proposed techniques make the overall system less practical for real applications. For example, the auto-regressive generation based on feature-dependency requires training multiple copies of the same pre-trained model;

2. It is unclear how the authors define/choose different task specific feature dependencies. Is it done randomly or using some domain specific knowledge? How will it be chosen for a brand new task? There are no experiments on the how sensitive is the performance on the choice of such feature dependency;

3. Many key details are missing:  What is meant by using KnowDA as task solver? Does it mean the pre-trained seq-to-seq model is directly being finetuned on the sequence labeling task? If so, is this model different than the one being used to generated the augmentation samples?
How many exemplars are chosen (per sample) for each task during pre-training and fine-tuning? Is one exemplar sufficient? If more than one is needed then how sensitive is the model to this choice?

Table 4 and its description on page 6 is not clear.
Figure 3 caption: Please add what is meant by T and D. Figure captions should be self-contained and understandable.

Typos:

Page 2: “.. KoMT is more scalable and comprehensive because those works heavily reply on the human-crafted prompts ..” -> heavily rely on
Page 2: “To summarize, contributes of this paper are following” -> contributions of this paper
Page 6: “We conduct the shot-10 setting where 40 samples for CoNLL’03 and 30 samples for WikiAnn”  -> setting with 40 samples

**Summary Of The Paper:**

This paper introduces a novel data augmentation model and technique for improving performance of few shot and low resource NLP tasks. Based on a seq-to-seq pre-trained language model (T5 large), this paper first present a multi-task pre-finetuning on 100+ NLP task using a new Knowledge Mixture Training (KoMT) framework, where all the datasets are represented in an unified  key-value pair format, adding some demonstration exemplars at the beginning, and training using a de-noising objective. For augmenting a low resource NLP task, the new samples are generated autoregressively according to task specific feature dependency. To accomplish this, different copies of the pre-finetuned model are trained for each auto-regressive step. The proposed method is evaluated on benchmark low resource NLP datasets (FewGLUE, CoNLL’03, and WikiAnn) and shown improvements over previous SOTA baselines.

**Summary Of The Review:**

The paper develops a novel multi-task pre-training technique for large language models, trained over 100+ NLP tasks, which can be subsequently used for data augmentation in a new unseen low resource NLP task. The paper demonstrates improved performance of the proposed data augmentation method on benchmark low resource NLP datasets. These new findings could be beneficial to the community.

---

### Decision · Program_Chairs · 2023-01-20

**Decision:**

Accept: poster

**Justification For Why Not Higher Score:**

One additional baseline that appears to be missing is a comparison with multitask fine-tuning, which is another way to improve low resource performance when given access to additional datasets. The approach also requires per-task work to define its key/value format, which may limit scalability to thousands of tasks.

**Justification For Why Not Lower Score:**

The reviewers agree that the method is thoroughly evaluated and achieves impressive results compared to strong baselines, and that the paper is clearly written. The authors also promise to share their code and model. Overall the paper is a useful contribution and is ready for acceptance at ICLR.


**Metareview: Summary, Strengths And Weaknesses:**

The paper show how to improve data augmentation for low resource NLP tasks by (1) unifying multiple tasks in the same data format, (2) training a single T5 model to generate data for over 100 NLP tasks, and (3) generating new data for previously unseen tasks.

The reviewers agree that the method is thoroughly evaluated and achieves impressive results compared to strong baselines, and that the paper is clearly written. The authors also promise to share their code and model.

One additional baseline that appears to be missing is a comparison with multitask fine-tuning, which is another way to improve low resource performance when given access to additional datasets. The approach also requires per-task work to define its key/value format, which may limit scalability to thousands of tasks.

Overall the paper is a useful contribution and is ready for acceptance at ICLR.

**Note From Pc:**

if the above contains the word "oral" or "spotlight" please see: "oral" presentation means -> notable-top-5% and "spotlight" means -> notable-top-25%. As stated in our emails, we are disassociating presentation type from AC recommendations

**Summary Of Ac-Reviewer Meeting:**

n/a